# Optimization of Fermentation and Biocontrol Efficacy of *Bacillus atrophaeus* XHG-1-3m2

**DOI:** 10.3390/microorganisms12112134

**Published:** 2024-10-24

**Authors:** Ziyan Xu, Hailong Lu, Wanbin Shi, Xinmei Zhou, Jianxin Ren, Yanling Zhang, Rong Ma

**Affiliations:** 1College of Forestry and Horticulture, Xinjiang Agricultural University, Urumqi 830052, China; m15609933188@163.com (Z.X.); luhailong0908@163.com (H.L.); 18097723209@163.com (W.S.); 13699955465@163.com (X.Z.); 2Yili State Forestry Pest Control and Quarantine Bureau, Ili 835000, China; ylzsfz@163.com; 3Xinyuan County Forestry and Grassland Bureau, Ili 835800, China; zy36589@163.com

**Keywords:** *Bacillus atrophaeus* XHG-1-3m2, liquid fermentation, optimization, activity assay, biocontrol

## Abstract

Biological control plays an increasingly important role in various aspects of modern agriculture and forestry. Identifying biocontrol strains with commercial potential for effective disease management is currently a focal point in biological control research. In this study, *Bacillus atrophaeus* XHG-1-3m2, a strain with significant biocontrol potential against *Wilsonomyces carpophilus* causing shot hole disease in wild apricots, was developed. The study determined the antibacterial activity of the fermentation broth, the optimal fermentation medium composition and conditions, and explored its effectiveness in controlling *Wilsonomyces carpophilus*. The optimal fermentation medium for strain XHG-1-3m2 comprises 12.5 g/L yeast extract, 12.5 g/L soy peptone, 10.0 g/L sodium chloride, 1 g/L ammonium chloride, 1 g/L potassium dihydrogen phosphate, 1 g/L disodium hydrogen phosphate, and 0.5 g/L magnesium sulfate heptahydrate. With an initial pH of 7.0, a liquid volume of 40%, an inoculum volume of 3%, and shaking incubation at 28 °C for 24 h, the viable cell count reached 14 × 10^9^ CFU/mL. In vitro and in vivo tests on leaves revealed that the fermentation broth and the biocontrol biofertilizer derived from this strain inhibited the leaf lesions caused by *Wilsonomyces carpophilus* on wild apricots, achieving inhibition rates of 94.62% and 82.46%, respectively.

## 1. Introduction

The Tianshan wild fruit forests in the Ili region of Xinjiang are an important germplasm gene bank for studying the genetic diversity and gene evolution of temperate fruit trees globally. The development and protection of these ecological resources are crucial [1]. In recent years, the fungal shot hole disease of wild apricots caused by *Wilsonomyces carpophilus* has occurred frequently [2], posing a significant risk to the germplasm resources of wild apricots and the stability of the ecosystem [3]. Therefore, in the context of the sustainable utilization and conservation of the rare and relict broadleaf forest ecosystems in the wild fruit forests of Tianshan, studying the green control of *Wilsonomyces carpophilus* and its damage to wild apricots has both practical and scientific significance.

Biological control is gaining widespread attention in modern green control systems due to its effectiveness in avoiding environmental pollution and resistance development in pathogens [4,5]. *Bacillus* spp. has been extensively studied for its antagonistic potential in controlling various plant diseases. As early as 1991, researchers evaluated the complete inhibition of aflatoxin synthesis by iturin A in the metabolites of *Bacillus subtilis* NK330, which lead to a better antagonistic effect of this strain against *Aspergillus flavus* and *Aspergillus parasiticus* [6]. Subsequently, numerous studies have demonstrated that the application of *Bacillus* strains or other antagonistic formulations in the field yields positive results [7,8,9]. Additionally, *Bacillus* spp. has shown specificity in controlling certain plant diseases. Most of these strains can effectively control diseases by producing antimicrobial substances, such as surfactins, iturins, and fengycins in the lipopeptides, as well as bacteriocins, polyketides, and terpenoids, etc. [10]. Thus, using *Bacillus* spp. as a biological control agent against various plant diseases holds tremendous potential.

The fermentation of antagonistic strains serves as a bridge between laboratory research and field applications of biocontrol agents. The optimization of fermentation aims to increase the number of viable cells or the content of antibacterial metabolites in the fermentation broth, ensuring the efficacy of biocontrol agents against plant diseases [11]. The efficacy of biocontrol agents is influenced not only by the intrinsic properties of the strains but also by the fermentation medium and conditions. Therefore, optimizing the fermentation medium and conditions is a prerequisite for the commercial production of biocontrol agents [12]. Currently, microbial fermentation optimization primarily employs single-factor, orthogonal, and response surface methodologies [13]. For instance, Zhong et al. [14] optimized the fermentation medium and conditions for the biocontrol bacterium FDHY-MZ2, increasing the bacterial biomass by 10.92 times. Dai et al. [15] reported an 87.04% inhibition rate of *Sphaeropsis sapinea* growth using the fermentation filtrate of *Bacillus pumilus* HR10 cultivated in an optimized medium. Ahsan et al. [16] identified and fermented the antagonistic strain *Bacillus amyloliquefaciens* BAM (TL-6), which holds promise for agricultural applications. Ahmady [17] investigated the fermentation of antagonistic bacteria in various liquid media. Additionally, Li et al. [18] studied the optimization of fermentation conditions for antagonistic bacteria targeting specific plant pathogens.

*Bacillus atrophaeus* XHG-1-3m2, isolated from infected wild apricot leaves in the Ili region of Xinjiang, has shown promising biocontrol potential against shot hole disease in wild apricots [19]. Due to Xinjiang’s unique geographic location and climatic characteristics, as well as the distinct biological traits of the microorganisms, there are significant challenges in applying biocontrol agents in this region. To better develop and utilize this biocontrol agent, this study optimized the fermentation medium and conditions for *Bacillus atrophaeus* XHG-1-3m2. The optimized fermentation broth was further developed into a biofertilizer, and its efficacy in controlling shot hole disease in wild apricots was evaluated through both in vitro and in vivo assays. The aim is to provide technical parameters and a theoretical basis for the further development and application of this biocontrol agent.

## 2. Materials and Methods

### 2.1. Experimental Materials

Phytopathogenic bacteria: The *Wilsonomyces carpophilus* (strain NO. XJAU G052 5m2) was isolated from wild apricot leaves in the Tianshan wild fruit forests which caused the shot hole disease in wild apricots. Species identification was completed in a previous study [3].Antagonistic bacteria: The antagonistic strain *Bacillus atrophaeus* XHG-1-3m2 was isolated from infected wild apricot leaves collected in the Yili region of Xinjiang. The strain is preserved in the Forest Protection and Pathology Laboratory, College of Forestry and Landscape Architecture, Xinjiang Agricultural University [19].Host plant: 2–3-year-old wild apricot plants.

### 2.2. Experimental Medium

Potato Dextrose Agar (PDA) Medium: 200 g potatoes, 20 g glucose, 17 g agar, 1000 mL distilled water, natural pH [20].

Luria–Bertani (LB) Solid Medium: 10 g tryptone, 5 g yeast extract, 10 g NaCl, 17 g agar, 1000 mL distilled water, pH 7.0–7.2.

LB Liquid Seed Medium: 10 g tryptone, 5 g yeast extract, 10 g NaCl, 1000 mL distilled water, pH 7.0–7.2.

Fermentation Medium 1 (Modified LB Liquid Medium): 10 g tryptone, 5 g yeast extract, 10 g NaCl, 1 g NH_4_Cl, 1 g KH_2_PO_4_, 1 g Na_2_HPO_4_, 0.5 g MgSO_4_·7H_2_O, 1000 mL distilled water, pH 6.7–7.0.

Fermentation Medium 2 (Nutrient Broth, NB): 10 g peptone, 3 g beef extract, 5 g NaCl, 1000 mL distilled water, pH 7.4.

Fermentation Medium 3 (PDB Medium): 200 g potatoes, 20 g glucose, 1000 mL distilled water, natural pH [20].

Fermentation Medium 4 [21]: 30 g cornmeal, 30 g soybean powder, 4 g K_2_HPO_4_, 1.5 g MgSO_4_·7H_2_O.

Fermentation Medium 5 [20]: 3% cornmeal, 6.0% soybean meal, 0.3% KH_2_PO_4_, 1000 mL distilled water, pH 7.0.

Fermentation Medium 6 [20]: 15 g wheat bran, 5 g soybean peptone, 0.1 g MnSO_4_, 3 g CaCO_3_, 1000 mL distilled water, pH 7.0.

Fermentation Medium 7 [20]: 5 g maltose, 10 g yeast extract, 5 g NaCl, 1000 mL distilled water, pH 7.0.

Fermentation Medium 8 [22]: 22.64 g glucose, 11.93 g peptone, 1.86 g yeast extract, 3 g NH_4_Cl, 1 g KH_2_PO_4_, 1 g Na_2_HPO_4_, 0.5 g MgSO_4_·7H_2_O, 5 g soybean powder, 1000 mL distilled water, pH 7.0.

Fermentation Medium 9 [23]: 5 g glucose, 20 g cornmeal, 40 g soybean meal, 3 g K_2_HPO_4_, 1 g KH_2_PO_4_, 2 g (NH_4_)_2_SO_4_, 0.20 g MgSO_4_·7H_2_O, 0.020 g MnSO_4_, 1000 mL distilled water, pH 7.0.

All media were sterilized at 121 °C for 30 min in an autoclave before use.

### 2.3. Antagonistic Activity Test

#### 2.3.1. Preparation of Seed Culture, Fermentation Broth, and Sterile Filtrate

After reactivating the *Wilsonomyces carpophilus* pathogen stored at −4 °C, it was placed in the center of a PDA plate and incubated at a constant temperature of 25 °C in the dark for 7–14 days until use. *Bacillus atrophaeus* XHG-1-3m2, stored at −70 °C, was streaked onto an LB solid medium with an inoculation needle for reactivation. The inoculated LB plates were incubated upside down in a 37 °C incubator for 24 h in the dark. A single colony from the LB plate was picked with an inoculation loop and transferred to a 250 mL Erlenmeyer flask containing 100 mL LB liquid medium, which was incubated at 30 °C and 180 r/min for 24 h in a shaking incubator to obtain the seed culture of *Bacillus atrophaeus* XHG-1-3m2. Under single-factor variable conditions, the seed culture was inoculated into the optimized fermentation medium at a 2% inoculation volume and incubated by shaking for 24 h to produce the fermentation broth. The fermentation broth of strain XHG-1-3m2 was filtered through a 0.22 μm microporous membrane to obtain the sterile filtrate.

#### 2.3.2. Determination of the Inhibitory Activity of Fermentation Broth Against Pathogenic Mycelial Growth

To further investigate the inhibitory properties of strain XHG-1-3m2, the following experiments were conducted to assess the antibacterial activity of the fermentation broth, sterile filtrate, and volatile compounds produced by the strain. A modified confrontation method using fermentation broth on plates was employed [24], with uninoculated medium serving as the control. Each treatment was repeated three times, and the plates were incubated at 28 °C. Once the diameter of the control colonies reached 85% of the plate or covered the entire plate, the inhibition rate of the mycelial growth was calculated.
Inhibition Rate (%) = [(Colony Diameter of Control Group − Colony Diameter of Treatment Group)/Colony Diameter of Control Group] × 100%

#### 2.3.3. Measurement of Inhibitory Rate of Volatile Compounds

The inhibitory activity of the volatile compounds produced by *Bacillus atrophaeus* XHG-1-3m2 was determined using the double-dish inverted method, with modifications based on Zhao et al. [25]. Uninoculated medium was used as the control, and each treatment was repeated three times. The plates were incubated at a constant temperature of 28 °C. Once the control plates were fully colonized, the colony diameters were measured, and the inhibition rate was calculated using the above formula.

### 2.4. Determination of the Growth Curve of the Antagonistic Strain

One milliliter of the prepared seed culture was inoculated into fresh LB liquid medium and incubated in a shaking incubator at 30 °C, 180 r/min. Measurements were taken every 4 h for the first 24 h and then every 12 h afterward. The optical density (OD) at 600 nm and the viable cell count were measured to determine the growth curve of the antagonistic strain. The viable cell count was determined using the plate dilution method.

The seed culture was inoculated into different liquid fermentation media and incubated in a shaking incubator under constant conditions. The optical density (OD_600_) of the fermentation broth was measured using a spectrophotometer at 600 nm, with uninoculated fermentation medium as the control. Higher OD_600_ values indicate higher biomass, while lower OD_600_ values indicate lower biomass. The viable cell count in the fermentation broth was determined using the plate dilution method [26]. Each gradient was repeated in parallel three times, and after 24 h of incubation at 30 °C in an inverted position, the colonies were counted. The viable cell count (CFU: Colony-Forming Units) was calculated as:Viable Cell Count (CFU/mL) = Number of colonies on plate × dilution factor/volume of plated inoculum (mL)

### 2.5. Screening of Fermentation Media

#### 2.5.1. Screening of Basic Fermentation Media

The basic fermentation medium was selected from nine initial fermentation media (as described in Section 2.2). The medium pH was set to 7.0, and the liquid volume in the shaking flask was 100 mL in a 250 mL flask. The seed culture of *Bacillus atrophaeus* XHG-1-3m2 was inoculated at a 2% (*v*/*v*) inoculum size and incubated at 30 °C, 180 r/min in a shaking incubator for 24 h. The biomass and viable cell count in the fermentation broth were measured, and the medium with the highest biomass and viable cell count was selected as the basic fermentation medium for *Bacillus atrophaeus* XHG-1-3m2.

#### 2.5.2. Screening of Fermentation Medium Components

The optimal carbon sources (yeast extract, glucose, sucrose, maltose, cornmeal), nitrogen sources (tryptone, peptone, soy peptone, yeast extract, potassium nitrate), and inorganic salts (sodium chloride, potassium chloride, calcium carbonate, manganese sulfate, copper sulfate) for *Bacillus atrophaeus* XHG-1-3m2 were determined using a single-factor analysis method. The biomass and viable cell count were measured according to the methods in Section 2.4.

#### 2.5.3. Screening of Optimal Concentrations for Fermentation Medium Components

Using the optimal fermentation medium from Section 2.5.1 as the base, the original carbon source concentration was replaced with concentrations of 2.5, 5.0, 7.5, 10, 12.5, and 15 g/L; the nitrogen source concentrations were replaced with 7.5, 10, 12.5, 15, 17.5, and 20 g/L; and the inorganic salt concentrations were replaced with 2.5, 5.0, 7.5, 10, 12.5, and 15 g/L. The fermentation of *Bacillus atrophaeus* XHG-1-3m2 was carried out under the conditions described in Section 2.5.1. The concentration yielding the best results was selected as the optimal component concentration for *Bacillus atrophaeus* XHG-1-3m2 fermentation.

#### 2.5.4. Orthogonal Screening of Fermentation Medium Component Ratios

Based on the results of single-factor experiments, an orthogonal experiment was designed to further determine the optimal fermentation medium formulation by testing the optimal medium components selected in Section 2.5.2 and Section 2.5.3.

### 2.6. Optimization of Fermentation Conditions

#### 2.6.1. Single-Factor Screening of Fermentation Conditions

The optimal fermentation volume (50, 70, 100, 120, 150 mL), inoculum size (1%, 2%, 3%, 4%, 5%), initial pH (5, 6, 7, 8, 9), temperature (24, 27, 30, 33, 36 °C), and shaking speed (100, 120, 150, 180, 200 r/min) for *Bacillus atrophaeus* XHG-1-3m2 were determined using a single-factor analysis method. The biomass and viable cell count were measured according to the methods in Section 2.4.

#### 2.6.2. Orthogonal Experiment for Fermentation Volume, Inoculum Size, Initial pH, Temperature, and Shaking Speed

Based on the analysis of single-factor experiments, it was found that fermentation volume, initial pH, and temperature significantly influenced the growth of the antagonistic strain during shaking flask fermentation. Therefore, an L_9_ (3^3^) orthogonal experiment was designed to determine the optimal fermentation conditions by testing the fermentation volume, initial pH, and temperature [14].

### 2.7. Preliminary Preparation of Bacillus atrophaeus XHG-1-3m2 Biofertilizer

Using calcium carbonate as the carrier, the fermentation broth was mixed with the carrier in a 1:1 ratio by mass (m = 1:1) to prepare *Bacillus atrophaeus* XHG-1-3m2 powder. The powder was air-dried in a cool, ventilated place. After drying, the powder was mixed with an organic water-soluble fertilizer (Si Talin, containing amino acids and trace elements) at a mass ratio of 100:0.3 to produce biofertilizer [27,28]. A 50% carbendazim wettable powder treatment (Sichuan Runer Science and Technology Company, Mianzhu, China) was added to the biocontrol effect test concurrently to compare it with the biocontrol product. This treatment has a stronger inhibitory impact on *Wilsonomyces carpophilus* [19]. A certain amount of the biofertilizer was dissolved in water, diluted, and the biomass and viable cell count were determined using the same method.

### 2.8. Determination of Biocontrol Efficacy

The biocontrol efficacy of the antagonistic strain fermentation broth was determined in the experiments using a preventive protection method and a therapeutic method. Outdoors under completely natural conditions (the average environmental temperature was 21.17 °C, with an average relative humidity of 53.63%), the efficacy was evaluated by inoculating detached and live wild apricot leaves [29]. When selecting leaf samples, it is important to maintain a balance between new and old leaves, avoiding excessive differentiation, while adhering to a fully random sampling method.

Preventive (Protection) Method: Fresh wild apricot leaves were cut using sterilized scissors, and the leaves were punctured with sterilized needles. Different fermentation broths were sprayed until the liquid dripped from the leaves [30]. This process was repeated three times at 3 h intervals. After 24 h, a 5 mm-diameter pathogen plug from a 7-day-old culture was inoculated at the wound sites, and the leaves were wrapped in sterile wet cotton and plastic wrap. Leaves inoculated only with the pathogen served as the control, and each treatment was repeated with three times (*n* = 3). After 24 h of incubation at room temperature, the pathogen plugs, plastic wrap, and sterile cotton were removed from the leaves. Disease lesion area changes were observed daily for approximately 9 days, and the inhibition rate was calculated. The in vivo experiment followed the same method, except the leaves were not cut, and the treated live plant leaves were observed for changes in lesion area to calculate the inhibition rate.

Therapeutic Method: Fresh wild apricot leaves were cut using sterilized scissors, and the leaves were punctured with sterilized needles. A 5 mm-diameter pathogen plug from a 7-day-old culture was inoculated at the wound sites, and the leaves were wrapped in sterile wet cotton and plastic wrap. Leaves inoculated only with the pathogen served as the control, and each treatment was repeated three times (*n* = 3). After 24 h of incubation at room temperature, the pathogen plugs, plastic wrap, and sterile cotton were removed. Various fermentation broths were then sprayed onto the leaves until runoff occurred. This spraying process was repeated three times at 3 h intervals, followed by a humidification treatment using wet cotton. Disease lesion area changes were observed daily for approximately 9 days, and the inhibition rate was calculated. The in vivo experiment followed the same method, except the leaves were not cut, and the treated live plant leaves were observed for changes in lesion area to calculate the inhibition rate.

### 2.9. Data Analysis

Experimental data were organized using Excel 2021 and analyzed using SPSS 26.0 for independent *t*-tests or one-way ANOVA. Before conducting the main statistical tests, a Shapiro–Wilk test (*p* < 0.05) was performed to assess the normality of the dataset, and homogeneity of variance was evaluated using Hartley’s test (*p* < 0.05). Consequently, Duncan’s new multiple range test was applied to determine the significance of differences, and figures were generated using Origin 2021.

## 3. Results

### 3.1. Determination of Antagonistic Activity and Inhibition Effect of the Antagonistic Strain

To learn more about this strain’s bacteriostatic characteristics, after fermenting *Bacillus atrophaeus* XHG-1-3m2, its fermentation broth, the supernatant of the fermentation broth, the 10-fold diluted fermentation broth, the 100-fold diluted fermentation broth, and the sterile filtrate all significantly inhibited the growth of *Wilsonomyces carpophilus* mycelium (Figure 1). Clear inhibition zones were observed on the confrontation culture plates, with the fermentation broth and supernatant of *Bacillus atrophaeus* XHG-1-3m2 showing the best inhibitory effects, achieving inhibition rates of 87.60% and 81.24%, respectively. The 10-fold and 100-fold diluted fermentation broths, as well as the sterile filtrate, showed slightly lower inhibition rates of 78.28%, 74.14%, and 73.35%, respectively. The sterilized fermentation broth showed no inhibitory effect (Table 1).

The results of the inhibitory activity test of volatiles produced by *Bacillus atrophaeus* XHG-1-3m2 showed that the volatiles generated from the fermentation broth of XHG-1-3m2 could significantly inhibit the growth of *Wilsonomyces carpophilus* mycelium (Figure 2), with inhibition rates ranging from 25.16% to 61.58%. The experiment found that the volatiles produced by the fermentation broth and the supernatant after centrifugation had the highest inhibitory rates, at 61.58% and 59.78%, respectively. The 10-fold diluted fermentation broth had a moderate inhibition rate of 35.11%, while the 100-fold diluted fermentation broth had a lower inhibition rate of only 25.16% (Table 1).

### 3.2. Determination of the Growth Curve of the Strain

The concentration of *Bacillus atrophaeus* XHG-1-3m2 in the culture followed an “S”-shaped curve over time (Figure 3). From 0 to 2 h, the bacterial concentration was relatively low and showed almost no change, representing the lag phase of bacterial growth. From 4 h onwards, the bacterial growth rate began to increase, and the concentration of bacteria gradually increased with time. Therefore, 4 to 24 h marked the logarithmic growth phase, during which the bacteria grew at the fastest rate. The period from 24 to 48 h was the stationary phase, where bacterial growth slowed down, and the bacterial concentration tended to stabilize. At 48 h, the bacterial concentration of *Bacillus atrophaeus* XHG-1-3m2 reached its peak, with an OD_600_ value of 1.88, corresponding to a viable cell count of 28 × 10^7^ CFU/mL. After 48 h, the bacterial concentration began to gradually decrease, marking the decline phase of growth. Therefore, from the perspective of viable cell count, it is recommended to select 24 h-old cultures as the optimal seed culture age.

### 3.3. Fermentation Optimization

#### 3.3.1. Optimal Fermentation Medium

Among the nine tested media, *Bacillus atrophaeus* XHG-1-3m2 exhibited the best growth in Test Medium 1, with an OD_600_ value of 1.75 and a viable cell count of 28 × 10^7^ CFU/mL. This was followed by Test Media 6 and 7, with OD_600_ values of 1.6 and 1.4, and viable cell counts of 21 × 10^7^ CFU/mL and 19 × 10^7^ CFU/mL, respectively. The growth of *Bacillus atrophaeus* XHG-1-3m2 was poor in Test Media 3 and 4, with OD_600_ values of 0.19 and 0.01, and viable cell counts of 8 × 10^7^ CFU/mL and 6 × 10^7^ CFU/mL, respectively. The large discrepancies in OD_600_ values and viable cell counts between Test Media 3, 4, LB medium, and Test Media 6, 7 indicated that different fermentation media had a significant effect on the growth of *Bacillus atrophaeus* XHG-1-3m2 (Figure 4g). Therefore, Test Medium 1 was selected as the basic fermentation medium for *Bacillus atrophaeus* XHG-1-3m2.

#### 3.3.2. Screening of Fermentation Medium Components and Optimal Concentrations

The effects of different fermentation medium components on the growth of *Bacillus atrophaeus* XHG-1-3m2 were investigated, and the results are shown in Figure 4. Carbon sources had a significant effect on the growth of *Bacillus atrophaeus* XHG-1-3m2, with yeast extract being the most effective. The OD_600_ value reached 1.86, and the viable cell count was 14 × 10^8^ CFU/mL, significantly higher than other carbon sources. This indicates that *Bacillus atrophaeus* XHG-1-3m2 utilizes yeast extract more efficiently as a carbon source. Therefore, yeast extract was selected as the carbon source for the fermentation medium of *Bacillus atrophaeus* XHG-1-3m2.

Similarly, nitrogen sources also had a significant impact on the growth of *Bacillus atrophaeus* XHG-1-3m2. The best fermentation results were obtained with soy peptone as the nitrogen source, with an OD_600_ value of 1.89 and a viable cell count of 18 × 10^8^ CFU/mL, followed by tryptone and peptone. The poorest results were observed when ammonium chloride was used as the nitrogen source, with an OD_600_ value of 0.369 and a viable cell count of only 5 × 10^8^ CFU/mL. Considering the effects, sources, and costs of tryptone, soy peptone, and peptone, soy peptone was chosen as the nitrogen source for the fermentation medium, as it is more conducive to the practical fermentation effect and production efficiency.

The effects of different inorganic salts on the fermentation of *Bacillus atrophaeus* XHG-1-3m2 were explored using sodium chloride, which was present in the basic fermentation medium. After 24 h of incubation, *Bacillus atrophaeus* XHG-1-3m2 exhibited the highest OD_600_ value and viable cell count in the medium with sodium chloride, significantly different from the other inorganic salts tested. The lowest OD_600_ values and viable cell counts were observed when calcium carbonate and copper sulfate were used as inorganic salts. Therefore, sodium chloride was selected as the inorganic salt for further optimization.

To assess the effect of fermentation medium components such as carbon sources, nitrogen sources, and inorganic salts on the growth rate of *Bacillus atrophaeus* XHG-1-3m2, the original concentrations of yeast extract, soy peptone, and sodium chloride in the LB medium were replaced with different concentrations to fully evaluate and optimize the LB medium formulation. As shown in the figure, the bacterial concentration in the fermentation broth increased initially and then decreased as the concentrations of yeast extract, soy peptone, and sodium chloride were raised. The optimal concentrations of yeast extract, soy peptone, and sodium chloride were 10, 10, and 12.5 g/L, respectively. Therefore, for the subsequent orthogonal experiment with three factors and three levels, the concentrations of yeast extract were set to 7.5, 10.0, and 12.5 g/L, soy peptone to 7.5, 10.0, and 12.5 g/L, and sodium chloride to 10.0, 12.5, and 15.0 g/L, which were considered reasonable.

#### 3.3.3. Optimization of Fermentation Medium Components via Orthogonal Experiment

Based on the results of the single-factor experiments, an orthogonal experiment was designed. The optimal medium components for the growth of *Bacillus atrophaeus* XHG-1-3m2, selected from Section 2.5.2 and Section 2.5.3, were subjected to orthogonal experimentation to further determine the optimal medium formulation (Table 2).

The effects of different concentrations of carbon sources, nitrogen sources, and inorganic salts on the fermentation of *Bacillus atrophaeus* XHG-1-3m2 were explored, and the results are shown in Figure 5. The orthogonal experiment results for carbon sources, nitrogen sources, and inorganic salts showed RB > RA > RC, indicating that changes in the nitrogen source (soy peptone) had the greatest impact on the growth rate of strain XHG-1-3m2, followed by yeast extract and sodium chloride. The optimal orthogonal combination was A3B3C1, which corresponds to the optimal medium formulation (optimized LB medium) consisting of 12.5 g/L yeast extract, 12.5 g/L soy peptone, and 10.0 g/L sodium chloride. The final optimized medium formulation includes 12.5 g/L yeast extract, 12.5 g/L soy peptone, 10.0 g/L sodium chloride, 1 g/L ammonium chloride, 1 g/L potassium dihydrogen phosphate, 1 g/L disodium hydrogen phosphate, and 0.5 g/L magnesium sulfate heptahydrate.

#### 3.3.4. Optimum Fermentation Conditions

*Bacillus atrophaeus* XHG-1-3m2 was cultured under various fermentation conditions, and the results indicated that agitation speed, inoculation volume, liquid volume, initial pH, and temperature all affected the biomass and viable cell count of the fermentation broth (Figure 6). As agitation speed increased, the bacterial concentration in the fermentation broth first increased and then decreased. The highest OD_600_ value and viable cell count were observed at 180 r/min, reaching 2.02 and 27 × 10^8^ CFU/mL, respectively. When the agitation speed was higher or lower than 180 r/min, the bacterial concentration decreased. Therefore, 180 r/min was selected as the optimal agitation speed for *Bacillus atrophaeus* XHG-1-3m2 (Figure 6d).

Different inoculation volumes had a minimal effect on the viable cell count of strain XHG-1-3m2, with the most similar values observed at 3% and 4% inoculation volumes, both averaging 29 × 10^8^ CFU/mL. This indicates that an inoculation volume between 3% and 4% is suitable for the growth of the strain. However, at lower (1%) or higher (5%) inoculation volumes, the viable cell count decreased. Therefore, a 3% inoculation volume was selected as the optimal inoculation volume for *Bacillus atrophaeus* XHG-1-3m2 (Figure 6e).

Liquid volume, initial pH, and temperature had a more significant effect on the biomass and viable cell count of the fermentation broth, all showing a trend in first increasing and then decreasing. The highest biomass was observed at a liquid volume of 100 mL, an initial pH of 7, and a temperature of 30 °C, with corresponding OD_600_ values of 2.05, 1.86, and 2.06, respectively (Figure 6b,c). Therefore, in subsequent orthogonal experiments, liquid volumes of 70, 100, and 120 mL, initial pH levels of 6.0, 7.0, and 8.0, and temperatures of 28, 30, and 32 °C were considered reasonable choices.

#### 3.3.5. Orthogonal Experiment for Fermentation Condition Optimization

Based on the analysis of single-factor experiment results, it was found that liquid volume, initial pH, and temperature had a significant impact on the growth rate of *Bacillus atrophaeus* XHG-1-3m2 during shaking flask fermentation. Therefore, an L_9_ (3^3^) orthogonal table was designed, and a three-factor, three-level orthogonal experiment was conducted to determine the optimal liquid volume, initial pH, and temperature for the growth of *Bacillus atrophaeus* XHG-1-3m2 based on the factors screened in Section 2.6.1 (Table 3).

The effects of different fermentation conditions on the fermentation of *Bacillus atrophaeus* XHG-1-3m2 were explored, and the results are shown in Figure 7. The orthogonal experiment results for liquid volume, initial pH, and temperature indicated RC > RA > RB, suggesting that temperature had the greatest impact on the growth rate of *Bacillus atrophaeus* XHG-1-3m2, followed by liquid volume and initial pH. The optimal combination was A2B2C1, meaning the best fermentation conditions were a liquid volume of 100 mL, an initial pH of 7.0, and a temperature of 28 °C.

### 3.4. Biocontrol Efficacy

To evaluate the effect of *Bacillus atrophaeus* XHG-1-3m2 in controlling *Wilsonomyces carpophilus* on wild apricot leaves, the biocontrol efficacy of different concentrations of the antagonistic strain fermentation broth was tested on detached wild apricot leaves with shot hole disease. Three days after inoculating *W. carpophilus* in the CK treatment, visible lesions appeared on wild apricot leaves. After nine days of pathogen inoculation under different treatments, untreated wild apricot leaves displayed severe shot hole disease symptoms (Figure 8). However, in the leaves treated with the fermentation broth of the antagonistic strain, the disease development was significantly restricted, and this effect was amplified by increasing concentrations of the fermentation broth. The clear inhibition of lesion development was observed with different treatments on wild apricot leaves (Figure 8).

As shown in Table 4, the preventive treatment was generally more effective than the therapeutic treatment in controlling wild apricot shot hole disease. Interestingly, in both the preventive and therapeutic treatments, the inhibition rate of lesion development increased with higher concentrations of the antagonistic strain fermentation broth. In the preventive treatment, when the fermentation broth of *Bacillus atrophaeus* XHG-1-3m2 was at a concentration of 10^9^ CFU/mL, the optimized fermentation broth, the initial biofertilizer derived from the fermentation broth, and the 1000-fold dilution of carbendazim had comparable effects, with lesion areas of 1.74 ± 0.40, 1.10 ± 0.43, and 0.62 ± 0.32 mm^2^, respectively, and the highest inhibition rates of 94.62%, 96.61%, and 98.06%. At a fermentation broth concentration of 10^5^ CFU/mL, the 100-fold diluted fermentation broth resulted in a lesion area of 5.70 ± 0.28 mm^2^, with the lowest inhibition rate of 82.35%. Although the therapeutic treatment also showed the highest inhibition rates (89~92%) with the optimized fermentation broth, carbendazim, and biofertilizer, the difference was not statistically significant (*p* < 0.05). However, the inhibition rates in the therapeutic treatment were about 5~6% lower compared to the preventive treatment. These findings indicate that the fermentation broth of *Bacillus atrophaeus* XHG-1-3m2 can inhibit the growth and penetration of the pathogen’s mycelium and effectively limit disease development.

To study the biocontrol efficacy of the fermentation broth of the antagonistic strain against shot hole disease in wild apricot, leaves inoculated only with the pathogen were used as the control, and monitoring was conducted over a 10-day period (Figure 9). During the experiment, we observed that the control treatment started to show symptoms and lesions on day 3 after inoculation with the pathogen. By day 9, the lesions had reached their maximum size and were accompanied by perforation symptoms. The results showed that there were clear differences between the leaves treated with the fermentation broth of the antagonistic strain and the control leaves. Similar to the results for detached leaves, the preventive method was significantly more effective than the therapeutic method. Additionally, the inhibition rate of lesion development increased with the concentration of the fermentation broth, and the lesion area on the leaves gradually decreased.

As shown in Table 5, the biocontrol efficacy on live leaves was generally lower than on detached leaves, but the overall trend was similar. When the concentration of the fermentation broth was 10^9^ CFU/mL and applied using the preventive method, the optimized fermentation broth, the initial biofertilizer derived from the strain, and the 1000-fold diluted carbendazim had the best effects, with lesion areas of 4.35 ± 0.28, 2.93 ± 0.70, and 2.13 ± 0.34 mm^2^, respectively, and the highest inhibition rates of 82.46%, 86.30%, and 90.07%. At a fermentation broth concentration of 10^5^ CFU/mL, the lesion area was the largest, and the inhibition rate was the lowest, with values of 9.42 ± 1.46 mm^2^ and 61.79%, respectively. In the therapeutic method, the lesion inhibition rate on the leaves was lower, with the optimized fermentation broth showing a 3.61% lower inhibition rate compared to the preventive method and the 100-fold diluted fermentation broth treatment being 1.18 times less effective. Therefore, in the biocontrol efficacy of the fermentation broth of the antagonistic strain against shot hole disease on live wild apricot leaves, the increase in the concentration of the fermentation broth significantly reduced the lesion expansion caused by the further infection of the leaves by *Wilsonomyces carpophilus*.

## 4. Discussion

The wild fruit forests in the Ili region of Xinjiang, China, are an important global distribution area for wild fruit forests and serve as a rare natural gene bank for fruit trees. They are also an important germplasm resource for the genetic diversity and gene evolution of temperate fruit trees. The green, non-polluting control of tree diseases and biological control are essential for the protection of these wild resources. Currently, the control of shot hole disease caused by *Wilsonomyces carpophilus* in this region is still limited to laboratory and chemical methods [19,31]. Exploring the inhibitory mechanisms of antagonistic bacteria and how to use their fermentation broths and metabolites to effectively control diseases is a hot topic in the field of biological control [32,33,34]. Zheng et al. [35] found that *Pantoea jilinensis* strain D25 and its fermentation broth effectively inhibited the growth and spore germination of *Botrytis cinerea* and altered its mycelial morphology, thereby preventing the spread of the pathogen. In the study by Siahmoshteh et al. [36], the sterile filtrate of *Bacillus subtilis* exhibited a growth inhibition rate of up to 92% against *Aspergillus* and showed excellent proteolytic activity. *Bacillus* species mainly inhibit pathogen growth through competition, antagonism, and the induction of systemic resistance in plants [37,38,39]. In this study, we found that the fermentation broth, supernatant, diluted fermentation broth, sterile filtrate, and volatile compounds produced by *Bacillus atrophaeus* XHG-1-3m2 could all effectively inhibit the growth of *Wilsonomyces carpophilus*, which is consistent with the findings of Vicente-Díez et al. [40] and Le et al. [41]. According to earlier research on this strain, *Bacillus atrophaeus* XHG-1-3m2 can successfully prevent *Wilsonomyces carpophilus* mycelium from growing normally, resulting in malformations, and can prevent spore generation or germination [42]. The bacteriostatic effect of the strain’s fermentation broth and its volatile gas in this study further confirmed that the strain could produce bacteriostatic active substances for biocontrol. These substances could be enzymes, bacteriocins, lipopeptides, antibiotics, or other antimicrobial proteins, among other things. In future research, we will identify these substances using metabolomics or analytical chemistry. Through their cells and metabolites, antagonistic bacteria may prevent the formation of infections, indicating that *Bacillus atrophaeus* XHG-1-3m2 has a significant potential for biological control and may be further improved.

Optimizing the microbial fermentation process is key to industrializing microbial fermentation products, ensuring their effectiveness as biopesticides. In the research and production of biological control agents, reducing costs is usually achieved by using inexpensive raw materials and improving the effectiveness of the fermentation broth. Enhancing fermentation efficiency requires technical expertise, and significant resources are often invested in exploring fermentation processes. Various experimental techniques and design methods have been applied to optimize fermentation processes [43,44]. However, due to the complexity of medium composition and fermentation conditions, relying solely on single-factor methods may not yield the expected results. This approach increases workload, lengthens the experimental cycle, and may lead to unreliable or incorrect conclusions due to variations in experimental conditions and batches [45]. Therefore, combining single-factor methods with orthogonal experiments allows for the rapid identification of optimal fermentation medium ratios. For example, Shi et al. [46] significantly increased the viable cell count of *Bacillus Velezensis* YH-18 through optimization and enhanced the antibacterial activity of the sterile filtrate. Dai et al. [15] optimized the fermentation conditions of *Bacillus pumilus* HR 10, achieving an inhibition rate of 87.04% against *Sphaeropsis sapinea*. Wu et al. [47] screened a low-cost medium and significantly improved the production of fibrinolysin by *Bacillus subtilis* WR 350 using single-factor and orthogonal experiments, reducing the cost of the optimized medium to just 8.5% of the initial cost. While orthogonal experiments may yield results similar to those obtained using single-factor methods, they offer advantages such as reasonable factor-level distribution, at lack of a need for repeated experiments to estimate errors, and higher accuracy, making the results more scientific and convincing [48]. A preliminary assessment of the fermentation medium and circumstances for *Bacillus atrophaeus* XHG-1-3m2 was carried out in this work using a combination of single-factor and orthogonal experiments. The results indicated that the optimal fermentation conditions for this strain were as follows: 12.5 g/L yeast extract, 12.5 g/L soy peptone, 10.0 g/L sodium chloride, 1 g/L ammonium chloride, 1 g/L potassium dihydrogen phosphate, 1 g/L disodium hydrogen phosphate, 0.5 g/L magnesium sulfate heptahydrate, an initial pH of 7.0, 40% liquid volume, 3% inoculum volume, and shaking incubation at 28 °C for 24 h. These fermentation conditions are similar to those found by Pan et al. [49], likely because this strain efficiently and rapidly absorbs organic nutrients such as yeast extract and peptone, indicating that fermentation optimization has a significant impact on the viable cell count of the strain. Additionally, we discovered that the use of soy peptones as the primary source of nitrogen is more commercially and economically feasible than the original tryptone while also producing better fermentation outcomes. Soy peptones are less expensive and easier to extract from plants. In Ma et al.’s [50] study, the fermentation conditions included wheat bran (35%), rice husks (40%), cornmeal (20%), soybean meal (15%), corn starch (1.5%), beef extract (2.5%), and magnesium sulfate (1.5%). Under these conditions, fermentation at 32 °C and pH 7.0 for 44 h yielded 4.577 × 10^13^ CFU/mL. Ahsan et al. [12] used coarse flour and peanut extract as the carbon and nitrogen sources for *Bacillus velezensis* BP-1, and the optimized LB medium achieved a 90% inhibition rate against *Pseudomonas aeruginosa*. Even though all of the aforementioned research concentrates on Bacillus species, their fermentation conditions which frequently use inexpensive agricultural raw materials differ greatly from those of this study. Similar materials were chosen in the early phases of this investigation, but the fermentation outcomes were disappointing. This disparity could result from variations in the target pathogen, strain characteristics, and origin.

The actual biocontrol efficacy of antagonistic strains and their related products is a critical step toward commercialization and large-scale application. Fermentation broths and products of antagonistic strains have been shown to inhibit the development of various diseases. For example, the fermentation broth of *Bacillus mojavensis* UTF-33 significantly inhibited the growth of rice blast, and leaf spot counts from fermentation filtrate-treated leaves were significantly lower than those treated with sterile water or LB, comparable to carbendazim [51]. This is consistent with the biocontrol pattern observed in this study using the fermentation broth and biofertilizer of *Bacillus atrophaeus* XHG-1-3m2, further demonstrating the trend in antagonistic bacterial products replacing chemical pesticides. This provides feasibility for the commercialization and large-scale field application of this strain. In another study, the fermentation broths of three antagonistic strains showed biocontrol efficacy against root rot in *Codonopsis*, with a disease inhibition rate of 59.24% using a fermentation broth concentration of 1 × 10^9^ CFU/mL [26], which is consistent with the concentration used in this study for biocontrol biofertilizer. In fact, our study revealed that the application method and concentration of biocontrol products, when used in conjunction with antagonists for biological control, can significantly impact the actual effectiveness of plant disease control. For this reason, we standardized the application method and included the concentration of each treatment group to minimize the impact in this area. Furthermore, the screening of endophytic antagonists has shown promising results in controlling wheat stripe rust, with the fermentation broth achieving a control rate of 55.97% [52]. In the study by Sun et al. [53], strain LHS11 effectively antagonized *Sclerotinia sclerotiorum*, achieving an inhibition rate of 85.71%. In vivo experiments with a composite agent (LHS11 + FX2) achieved a control efficiency of 80.51%.

## 5. Conclusions

This investigation verified that the pathogen *Wilsonomyces carpophilus*, which causes shot hole disease in wild apricots, is inhibited by *Bacillus atrophaeus* XHG-1-3m2. The biofertilizer derived from *Bacillus atrophaeus* XHG-1-3m2 fermentation broth demonstrated inhibition rates of 94.62% and 82.46%, respectively, against *Wilsonomyces carpophilus*-caused shot hole disease lesions on detached and living wild apricot leaves. The employment of antagonistic microorganisms for environmentally friendly, non-polluting biological management of plant diseases is consistent with the ecological balance and environmental protection tenets promoted by international environmental organizations. With this investigation, the shot hole disease on wild apricots caused by *Wilsonomyces carpophilus* was significantly reduced. The *Bacillus atrophaeus* XHG-1-3m2 fermentation formula and circumstances were not only optimized, but tests on detached and living leaves showed that the fermentation broth and its byproducts were also successful in managing the disease. The large-scale field application and commercial biocontrol agent development of *Bacillus atrophaeus* XHG-1-3m2 are made possible by its biocontrol efficacy. Investigating and characterizing the bacteriostatic metabolites of this strain and clarifying the biocontrol interactions between *Wilsonomyces carpophilus* and *Bacillus atrophaeus* XHG-1-3m2 by omics techniques should be the main foci of future exploration. To investigate the mechanism of interaction between pathogenic bacteria, antagonistic bacteria, and host plants, high-throughput multi-omics technology will be utilized in the interim.

## Figures and Tables

**Figure 1 microorganisms-12-02134-f001:**
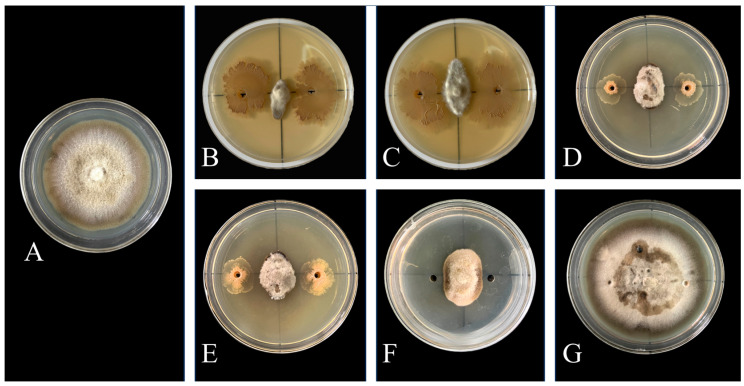
Inhibitory effects of different treated fermentation broths on the pathogen after 30 days of confrontation culture at 25 °C. Note: (**A**) CK (control); (**B**) fermentation broth; (**C**) supernatant after centrifugation; (**D**) fermentation broth diluted 10 times; (**E**) fermentation broth diluted 100 times; (**F**) sterile filtrate (supernatant filtered through a 0.22 μm membrane); (**G**) sterilized fermentation broth (autoclaved).

**Figure 2 microorganisms-12-02134-f002:**
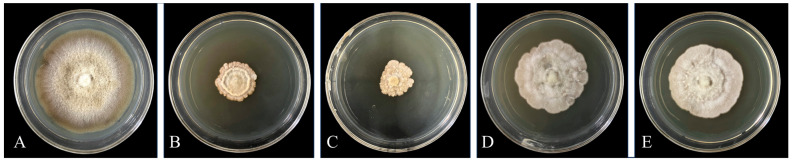
Inhibitory effects of volatile compounds from different treated fermentation broths on the pathogen after 30 days of confrontation culture at 25 °C. Note: (**A**) CK (control); (**B**) supernatant after centrifugation; (**C**) fermentation broth; (**D**) fermentation broth diluted 10 times; (**E**) fermentation broth diluted 100 times.

**Figure 3 microorganisms-12-02134-f003:**
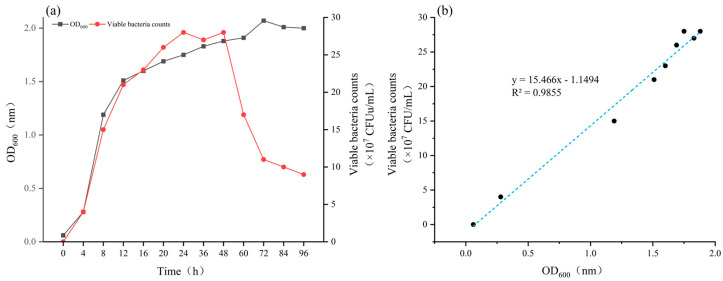
Growth curve of *Bacillus atrophaeus* XHG-1-3m2 (**a**) and the linear relationship between its viable cell count and OD_600_ (**b**).

**Figure 4 microorganisms-12-02134-f004:**
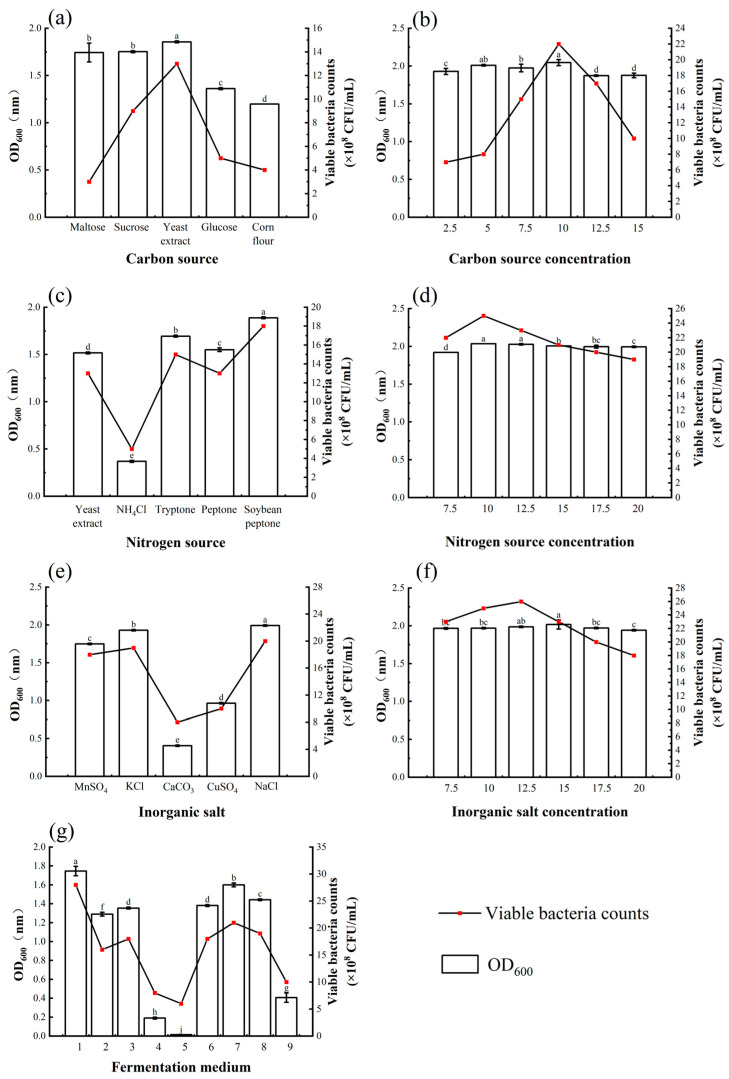
Effects of different fermentation medium, medium components, and their concentrations on the viable cell count and OD_600_ of *Bacillus atrophaeus* XHG-1-3m2. Note: (**a**) carbon sources; (**b**) carbon source concentration; (**c**) nitrogen sources; (**d**) nitrogen source concentration; (**e**) inorganic salts; (**f**) inorganic salt concentration; (**g**) fermentation medium. Different lowercase letters in the same column indicate a significant difference at *p* < 0.05 level as determined by a Duncan’s new multiple range test.

**Figure 5 microorganisms-12-02134-f005:**
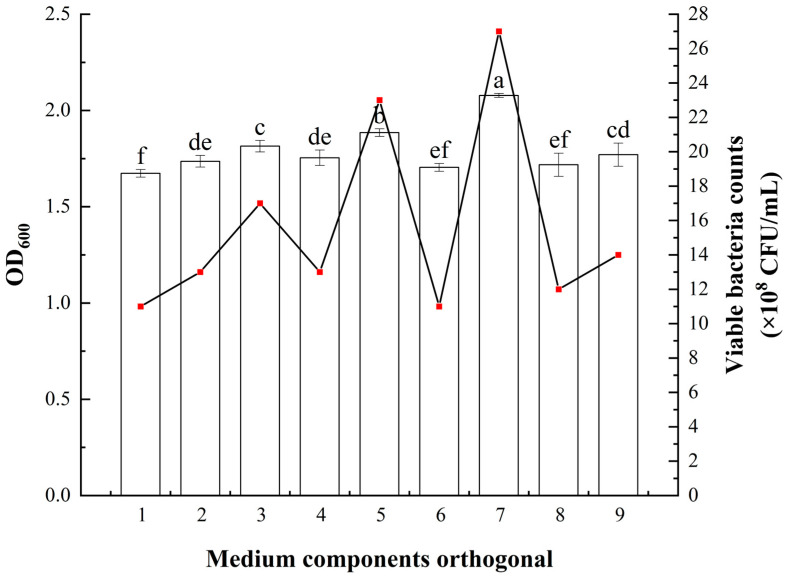
Effects of different fermentation medium components and their concentrations on the viable cell count and OD_600_ of *Bacillus atrophaeus* XHG-1-3m2. Note: Different lowercase letters in the same column indicate a significant difference at *p* < 0.05 level as determined by a Duncan’s new multiple range test.

**Figure 6 microorganisms-12-02134-f006:**
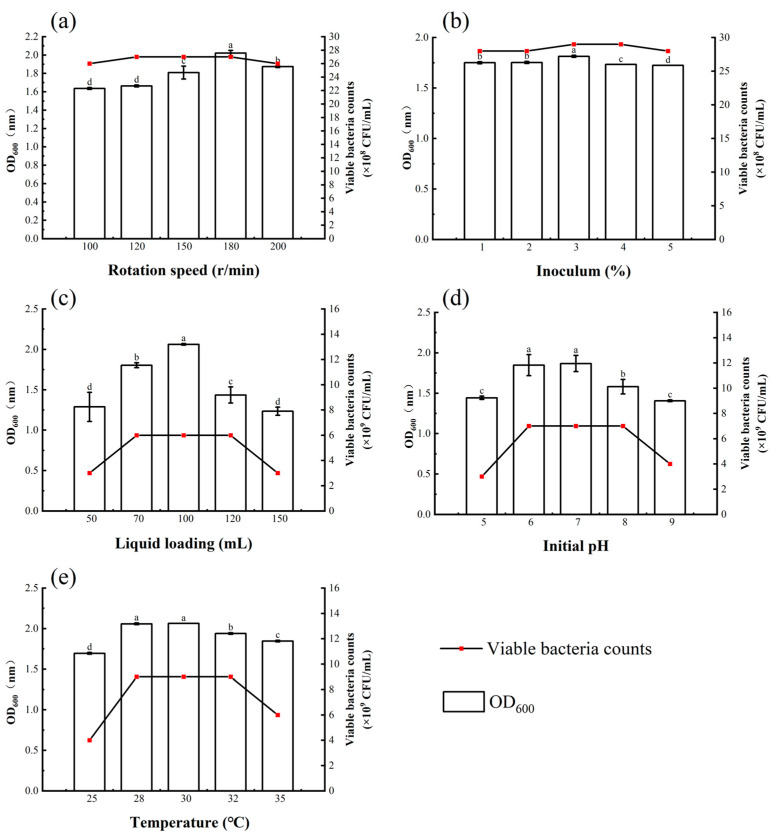
Effects of different fermentation conditions on the viable cell count and OD_600_ of *Bacillus atrophaeus* XHG-1-3m2. Note: (**a**) rotation speed; (**b**) inoculum; (**c**) liquid loading; (**d**) initial pH; (**e**) temperature. Different lowercase letters in the same column indicate a significant difference at *p* < 0.05 level as determined by a Duncan’s new multiple range test.

**Figure 7 microorganisms-12-02134-f007:**
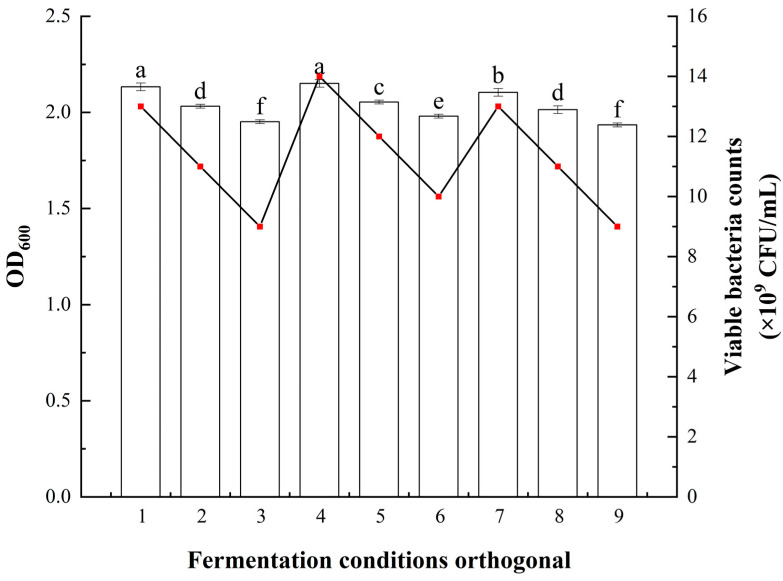
Effects of different fermentation conditions on the viable cell count and OD_600_ of *Bacillus atrophaeus* XHG-1-3m2. Note: Different lowercase letters in the same column indicate a significant difference at *p* < 0.05 level as determined by a Duncan’s new multiple range test.

**Figure 8 microorganisms-12-02134-f008:**
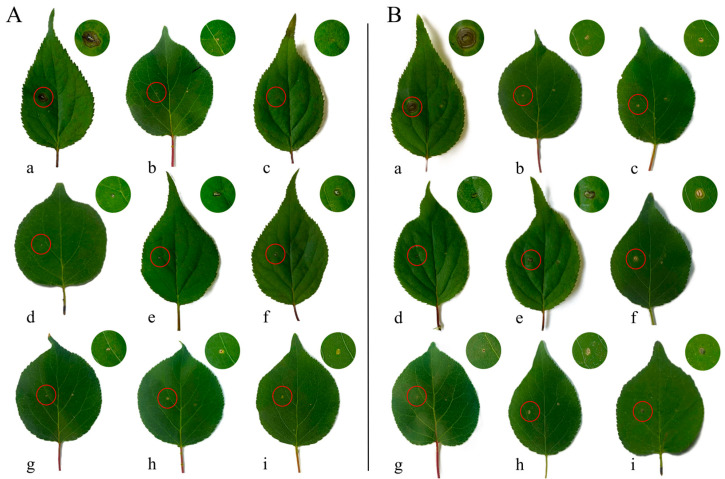
Biocontrol efficacy of different treated fermentation broths of the antagonistic strain on detached wild apricot leaves with shot hole disease using preventive (**A**) and therapeutic (**B**) methods. Note: a: CK; b: carbendazim; c: biofertilizer; d: fermentation broth; e: fermentation broth diluted 10 times; f: fermentation broth diluted 100 times; g: optimized fermentation broth; h: optimized fermentation broth diluted 10 times; i: optimized fermentation broth diluted 100 times.

**Figure 9 microorganisms-12-02134-f009:**
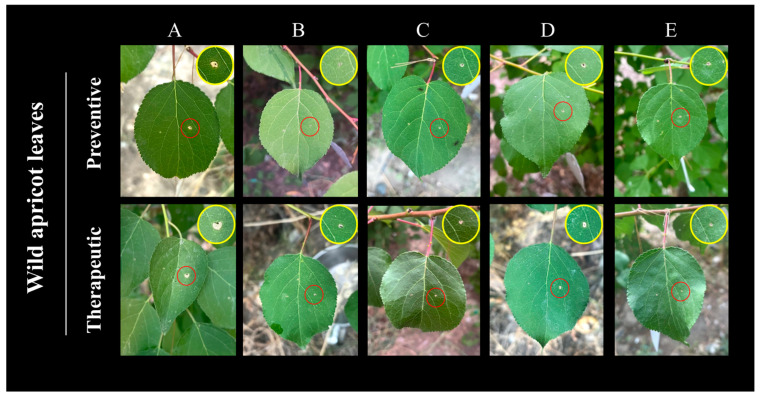
Biocontrol efficacy of different treated fermentation broths of the antagonistic strain on live wild apricot leaves with shot hole disease using preventive and therapeutic methods. Note: (**A**) CK; (**B**) carbendazim; (**C**) biofertilizer; (**D**) fermentation broth; (**E**) optimized fermentation broth.

**Table 1 microorganisms-12-02134-t001:** Effect of different treatments of fermentation broth with antagonistic strains of bacteria on the inhibitory activity of the bacteria.

Treatment	Mean Colony Diameter (mm)	Inhibition Rate (%)
CK	76.73 ± 1.99 a	-
Fermentation broth	9.51 ± 0.64 h	87.60 ± 0.95 a
Supernatant after centrifugation	14.40 ± 1.29 g	81.24 ± 1.58 b
Fermentation broth diluted 10 times	16.66 ± 0.97 f	78.28 ± 1.24 b
Fermentation broth diluted 100 times	19.84 ± 1.83 e	74.14 ± 2.36 c
Sterile filtrate (supernatant filtered through a 0.22 μm membrane)	20.44 ± 2.29 e	73.35 ± 3.11 c
Sterilized fermentation broth (autoclaved)	-	-
Volatiles from fermentation broth with live bacteria	29.44 ± 1.89 d	61.58 ± 3.15 d
Volatiles from centrifugation supernatant	30.86 ± 2.37 d	59.78 ± 2.86 d
Volatiles from 10-fold dilution of fermentation broth with live bacteria	49.75 ± 1.21 c	35.11 ± 3.02 e
Volatiles from 100-fold dilution of fermentation broth with live bacteria	57.41 ± 1.59 b	25.16 ± 1.99 f

Note: CK—control; “-” indicates no inhibitory effect. Different lowercase letters in the same column indicate a significant difference at *p* < 0.05 level as determined by a Duncan’s new multiple range test.

**Table 2 microorganisms-12-02134-t002:** Results of the orthogonal experiment on the optimal medium components and their concentrations for the antagonistic strain.

Treatment	Factor			OD_600_
	Yeast Extract (A)	Soybean Peptone (B)	NaCl (C)	
1	7.5	7.5	10	1.674
2	7.5	10	12.5	1.736
3	7.5	12.5	15	1.815
4	10	10	10	1.754
5	10	12.5	12.5	1.885
6	10	7.5	15	1.704
7	12.5	12.5	10	2.078
8	12.5	7.5	12.5	1.718
9	12.5	10	15	1.771
K1	5.224	5.096	5.506	
K2	5.344	5.261	5.339	
K3	5.567	5.778	5.290	
k1	1.741	1.699	1.835	
k2	1.781	1.754	1.780	
k3	1.856	1.926	1.763	
R	0.114	0.227	0.072	

Note: K represents the sum of three OD_600_ values at the same level as a certain factor, and k is the corresponding K/3. R represents the level difference between k1, k2, and k3 under the same factor. Higher values of R signify a pronounced impact on the test results, with the optimal level correlating to the factor yielding the highest value of k. RA: R of yeast extract, RB: R of soybean peptone, RC: R of NaCl.

**Table 3 microorganisms-12-02134-t003:** Results of the orthogonal experiment for optimal fermentation conditions of the antagonistic strain.

Treatment	Factor			OD_600_
	Liquid Loading (mL) (A)	Initial pH (B)	Temperature (°C) (C)	
1	70	6	28	2.133
2	70	7	30	2.032
3	70	8	32	1.952
4	100	7	28	2.151
5	100	8	30	2.054
6	100	6	32	1.980
7	120	8	28	2.104
8	120	6	30	2.014
9	120	7	32	1.935
K1	6.117	6.128	6.388	
K2	6.185	6.118	6.100	
K3	6.053	6.110	5.867	
k1	2.039	2.043	2.129	
k2	2.062	2.039	2.033	
k3	2.018	2.037	1.956	
R	0.044	0.006	0.174	

Note: K represents the sum of three OD_600_ values at the same level as a certain factor, and k is the corresponding K/3. R represents the level difference between k1, k2, and k3 under the same factor. Higher values of R signify a pronounced impact on the test results, with the optimal level correlating to the factor yielding the highest value of k. RA: R of liquid loading, RB: R of initial pH, RC: R of temperature.

**Table 4 microorganisms-12-02134-t004:** Biocontrol efficacy of different treated fermentation broths of the antagonistic strain on detached wild apricot leaves with shot hole disease using preventive and therapeutic methods.

Treatment	Methods and Indicators			
	Preventive		Therapeutic	
	Area of Disease Spots (mm^2^)	Inhibition Rate (%)	Area of Disease Spots (mm^2^)	Inhibition Rate (%)
CK	32.85 ± 5.26 a	-	38.83 ± 5.32 a	-
Fermentation broth (10^7^ CFU/mL)	3.64 ± 0.95 bcd	88.67 ± 3.79 cd	7.53 ± 1.69 bcd	80.51 ± 3.96 bc
Fermentation broth diluted 10 times (10^6^ CFU/mL)	4.47 ± 0.68 bc	86.27 ± 2.26 d	9.26 ± 0.92 bc	75.88 ± 3.91 cd
Fermentation broth diluted 100 times (10^5^ CFU/mL)	5.70 ± 0.28 b	82.35 ± 2.89 e	10.95 ± 0.47 b	71.53 ± 2.99 d
Optimized fermentation broth (10^9^ CFU/mL)	1.74 ± 0.40 cd	94.62 ± 1.52 ab	4.15 ± 0.72 de	89.12 ± 2.60 a
Optimized fermentation broth diluted 10 times (10^8^ CFU/mL)	2.62 ± 0.69 bcd	92.12 ± 1.20 bc	6.99 ± 1.15 cd	81.90 ± 2.66 b
Optimized fermentation broth diluted 100 times (10^7^ CFU/mL)	3.43 ± 0.51 bcd	89.35 ± 2.09 cd	8.27 ± 1.09 bc	78.69 ± 0.18 bc
Carbendazim (0.5 g/L)	1.10 ± 0.43 cd	96.61 ± 1.33 a	3.29 ± 0.21 e	91.43 ± 1.88 a
Biofertilizer (10^9^ CFU/mL)	0.62 ± 0.32 d	98.06 ± 1.01 a	2.96 ± 0.54 e	92.39 ± 0.67 a

Note: Different lowercase letters in the same column indicate a significant difference at *p* < 0.05 level as determined by a Duncan’s new multiple range test.

**Table 5 microorganisms-12-02134-t005:** Biocontrol efficacy of different treated fermentation broths of the antagonistic strain on live wild apricot leaves with shot hole disease using preventive and therapeutic methods.

Treatment	Methods and Indicators			
	Preventive		Therapeutic	
	Area of Disease Spots (mm^2^)	Inhibition Rate (%)	Area of Disease Spots (mm^2^)	Inhibition Rate (%)
CK	21.81 ± 2.76 a	-	24.92 ± 1.90 a	-
Fermentation broth (10^7^ CFU/mL)	5.51 ± 0.35 cd	77.77 ± 2.84 bcd	6.72 ± 0.71 bc	72.81 ± 4.70 cd
Fermentation broth diluted 10 times (10^6^ CFU/mL)	7.50 ± 1.24 bc	69.53 ± 7.24 def	7.40 ± 1.76 bc	70.50 ± 4.86 cd
Fermentation broth diluted 100 times (10^5^ CFU/mL)	9.42 ± 1.46 b	61.79 ± 8.70 f	8.26 ± 1.20 b	66.86 ± 4.28 d
Optimized fermentation broth (10^9^ CFU/mL)	4.35 ± 0.28 de	82.46 ± 1.76 abc	5.26 ± 0.55 cd	78.85 ± 1.87 bc
Optimized fermentation broth diluted 10 times (10^8^ CFU/mL)	6.21 ± 1.60 cd	74.71 ± 8.40 cde	6.57 ± 1.28 bc	73.55 ± 1.32 cd
Optimized fermentation broth diluted 100 times (10^7^ CFU/mL)	8.63 ± 0.57 b	65.26 ± 3.26 ef	7.75 ± 1.06 b	68.80 ± 4.55 d
Carbendazim (0.5 g/L)	2.93 ± 0.70 e	86.30 ± 4.21 ab	4.15 ± 1.05 d	83.11 ± 5.53 ab
Biofertilizer (10^9^ CFU/mL)	2.13 ± 0.34 e	90.07 ± 2.44 a	3.36 ± 0.33 d	86.53 ± 0.50 a

Note: Different lowercase letters in the same column indicate a significant difference at *p* < 0.05 level as determined by a Duncan’s new multiple range test.

## Data Availability

All data are included in the paper.

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
