# Peer review of "Optimization of Fermentation and Biocontrol Efficacy of *Bacillus atrophaeus* XHG-1-3m2"

_microorganisms, 2024, doi:10.3390/microorganisms12112134_

Round 1

Reviewer 1 Report

Comments and Suggestions for Authors

The paper "Optimization of Fermentation and Biocontrol Efficacy of Bacillus atrophaeus XHG-1-3m2" addresses an important issue in sustainable agricultureby optimizing fermentation processes for a biocontrol agent. The study shows promise in reducing the harmful effects of shot hole disease in wild apricots, caused by Wilsonomyces carpophilus.

Some suggestions for Authors for upgrading their work:

Suggestions for Improvement:

  1. Your paper explains the optimization processes for fermentation, but it could benefit from a stronger discussion of why specific media components were chosen and their cost-effectiveness for future scaling. A deeper exploration of the economic feasibility of the chosen media components would be helpful, especially when considering commercialization.
  2. After primary reading, I made an impression that the paper could improve by elaborating on the underlying mechanisms through which Bacillus atrophaeus XHG-1-3m2 inhibits the growth of Wilsonomyces carpophilus. You mention the inhibition effect, but further studies exploring how metabolites produced during fermentation impact pathogen suppression would strengthen the paper's contributions to the field.
  3. It would be great to compare Bacillus atrophaeus XHG-1-3m2 with other strains used for similar purposes in agriculture. This part is missing in this version.
  4. Some results, particularly those from fermentation optimization, could be made more accessible by providing clearer visualizations. For example, using more detailed graphs or additional tables to summarize the outcomes of the orthogonal experiments would enhance the clarity of the findings.
  5. The conclusion section is very bad; I think that could summarize the potential impacts of the study more effectively, including specific suggestions for further research, such as molecular-level investigations or the exploration of other pathogenic targets.

Reviewer 2 Report

Comments and Suggestions for Authors

A review report

This study investigates the biocontrol potential of Bacillus atrophaeus XHG-1-3m2, focusing on its effectiveness against Wilsonomyces carpophilus, which causes shot hole disease in wild apricots. The authors comprehensively evaluated the strain’s antibacterial properties, optimized the fermentation conditions for maximal growth, and assessed its efficacy in controlling the pathogen in both in vitro and in vivo settings. The results indicate promising applications for this strain in agricultural disease management.

Comments for authors

1.       The materials and methods section provides an overview of the experimental design but lacks comprehensive detail regarding randomization and replication protocols. Specific information about how treatments were allocated within experimental units and any controls employed is limited. For example, there is no mention of whether the allocation of treatments was done randomly across the entire experimental area or whether block designs were utilized to account for potential variability in environmental conditions.

2.       The methods used to introduce the Bacillus atrophaeus XHG-1-3m2 strain into the experimental setup are not adequately detailed. Understanding the inoculation procedure is crucial, as it influences the effectiveness of biocontrol. Specific parameters such as inoculum concentration, method of application (e.g., spraying, drenching), and timing relative to pathogen exposure should be elaborated upon.

3.       While the study briefly mentions the conditions under which experiments were conducted, it does not provide comprehensive data on other environmental factors, such as humidity and temperature variations during the study periods. These factors can significantly influence the behavior and effectiveness of both the biocontrol agent and the target pathogens.

4.       The description of statistical methods used to analyze the data is insufficient. Specific statistical tests applied, assumptions for these tests, and any adjustments made for multiple comparisons should be explicitly stated to enhance the reproducibility and interpretability of the results. The current outline does not clarify whether any preliminary data analysis was conducted prior to applying the main statistical tests.

5.       The section does not provide justification for the chosen sample sizes for both the biocontrol agent and the pathogen assessments. A larger sample size may be necessary to achieve reliable statistical power, particularly when testing multiple treatment groups and assessing variability within treatments.

By addressing these shortcomings, the Materials and Methods section would provide a more robust framework for the research, enhancing both its scientific rigor and reproducibility. This would ultimately contribute to a deeper understanding of the potential applications of Bacillus atrophaeus in sustainable agriculture. I recommend a major revision.

Reviewer 3 Report

Comments and Suggestions for Authors

The peer-reviewed article is an original study devoted to the evaluation of the biocontrol properties of Bacillus atrophaeus XHG-1-3m2, as well as the optimization of the cultivation conditions of this strain. The authors have carried out a significant amount of experimental research. The conclusions given in the work do not cause doubts. Highly appreciating the work, I will note some remarks:

1. Based on the already published studies of other authors, the Introduction and Discussion should be rewritten in view of the information about the antagonistic properties of Bacillus atrophaeus in relation to other phytopathogenic microorganisms, and indicate which compounds may cause their biocontrol properties.

2. References to literary sources should be added to the Materials and methods section.

3. Some technical notes should be corrected (see article file).

Therefore, after thorough rewriting of the Introduction and Discussion, the article can be published.

Round 2

Reviewer 1 Report

Comments and Suggestions for Authors

/

Reviewer 2 Report

Comments and Suggestions for Authors

The authors have addressed all of my concerns, and I have no further objections. I recommend that the manuscript be accepted for publication.

Reviewer 3 Report

Comments and Suggestions for Authors

The authors have corrected all comments. The article contains the latest and important information. Can be published